# Doping strategy in metavalently bonded materials for advancing thermoelectric performance

Ming Liu[1,2], Muchun Guo[3], Haiyan Lyu[2], Yingda Lai[1], Yuke Zhu[1], Fengkai Guo [1] ✉, Yueyang Yang[2], Kuai Yu[1], Xingyan Dong[1], Zihang Liu [1], Wei Cai[1], Matthias Wuttig [2,4] ✉, Yuan Yu [2] ✉ & Jiehe Sui [1] ✉

Metavalent bonding is a unique bonding mechanism responsible for exceptional properties of materials used in thermoelectric, phase-change, and optoelectronic devices. For thermoelectrics, the desired performance of metavalently bonded materials can be tuned by doping foreign atoms. Incorporating dopants to form solid solutions or second phases is a crucial route to tailor the charge and phonon transport. Yet, it is difficult to predict if dopants will form a secondary phase or a solid solution, which hinders the tailoring of microstructures and material properties. Here, we propose that the solid solution is more easily formed between metavalently bonded solids, while precipitates prefer to exist in systems mixed by metavalently bonded and other bonding mechanisms. We demonstrate this in a metavalently bonded GeTe compound alloyed with different sulfides. We find that S can dissolve in the GeTe matrix when alloyed with metavalently bonded PbS. In contrast, S-rich second phases are omnipresent via alloying with covalently bonded GeS and SnS. Benefiting from the reduced phonon propagation and the optimized electrical transport properties upon doping PbS in GeTe, a high figure-of-merit $ZT$ of 2.2 at 773 K in $(Ge_{0.84}Sb_{0.06}Te_{0.9})(PbSe)_{0.05}(PbS)_{0.05}$ is realized. This strategy can be applied to other metavalently bonded materials to design properties beyond thermoelectrics.

Thermoelectric technology has significant application potential for waste heat harvesting and distributed cooling by directly converting heat into electricity and vice versa[1,2]. The conversion efficiency of thermoelectric materials is gauged by the dimensionless figure of merit $ZT = \alpha^2\sigma T/\kappa_{tot}$, where $\alpha$ is the Seebeck coefficient, $\sigma$ is the electrical conductivity, $T$ is the absolute temperature, and $\kappa_{tot}$ is the total thermal conductivity, including the electronic ($\kappa_e$) and lattice ($\kappa_L$) contributions[3,4]. To improve thermoelectric performance, strategies such as carrier concentration optimization[5–11], charge scattering mechanism regulation[12,13], electronic band structure manipulation[14–19], and micro/nanostructure modification[20–24], have been individually or synergistically adopted in different materials.

These strategies listed above are generally realized by doping foreign atoms and depend on the behavior of dopants in the matrix. In some cases, the foreign atoms can be uniformly distributed in the matrix, forming a solid solution. In this scenario, the carrier concentration, the energy band structure, and the scattering of high-frequency phonons can be effectively manipulated. Typical examples are $Bi_2Te_3$ doped with Sb or Se[25–27]; PbTe doped with Se, Sn, and Ge[14,28,29]; $Mg_2Si$ doped with Ge and Sn[30]; $Mg_3Sb_2$ doped with Bi[31–33];

[1]National Key Laboratory for Precision Hot Processing of Metals, Harbin Institute of Technology, Harbin, China. [2]Institute of Physics (IA), RWTH Aachen University, Aachen, Germany. [3]School of Materials Science and Engineering, Xihua University, Chengdu, China. [4]Green IT (PGI 10), Forschungszentrum Jülich GmbH, Jülich, Germany. ✉e-mail: fkguo@hit.edu.cn; wuttig@physik.rwth-aachen.de; yu@physik.rwth-aachen.de; suijiehe@hit.edu.cn

$CaMg_2Bi_2$ doped with Ba[17,18,34]; and NbFeSb doped with Zr, Hf and Ta[35,36]. In stark contrast, some foreign atoms show a very small solubility in the matrix, instead forming clusters[37,38], nanoprecipitates[39,40], and boundary complexions[41,42]. In this scenario, phonons in the intermediate and long wavelength range are more strongly scattered while maintaining a high electron mobility. For instance, Cd doping in $AgSbTe_2$[38] induces nanoscale superstructures. Nanoprecipitates are also found in Li-doped SnTe[43], Sr-doped PbTe[44,45], and Si-doped $CoSb_3$[46]. The different behavior of these dopants makes the optimization of thermoelectric properties via designing microstructures elusive because it is hard to predict how the dopants will behave in the host. Therefore, exploring a facile and feasible doping strategy by precisely adjusting the behavior of the dopants in the matrix will help to improve the thermoelectric properties.

From the viewpoint of solubility, chemical similarity is an important indicator of phase uniformity. For instance, for liquids, the polarity similarity of solute and solvent could favor a high solubility[47]; for solids, a small difference in the size and electronegativity between solute and solvent atoms is vital for forming a high solubility solid solution, as described by the Hume-Rothery rule[48]. In thermoelectric semiconductors, $Bi_2Te_3$ can form an infinite solid solution with $Sb_2Te_3$ or $Bi_2Se_3$[49]. In stark contrast, the solubility of $Bi_2S_3$ and $Sb_2Se_3$ in $Bi_2Te_3$ is much lower[23,50]. This raises an interesting and also important question: what mechanisms are responsible for these different solubilities even though these elements are close neighbors in the periodic table, i.e., they have very close atomic radii and electronegativity differences? Understanding this question will help us to better choose appropriate dopants to achieve the desired microstructures and properties.

Taking a typical thermoelectric material, GeTe, as an example, its thermoelectric performance enhancement strongly depends on the selection of dopants to realize the reduction of carrier concentration[51,52], the enhancement of valley degeneracy[53–56], and the minimization of lattice thermal conductivity[57,58]. Yet, until now, general and effective strategies have been mainly realized by substituting the cation sites with dopants such as Sb[52], Bi[6], and Pb[59]. Other dopants such as Ga[41] have a very low solubility in GeTe and thus form clusters and grain boundary complexions. By contrast, doping at the anion site has rarely been reported even though a few studies have proven it to be a promising approach to improve the thermoelectric performance of GeTe[60,61]. However, the low solubility of many anionic dopants in GeTe restricts the space for property manipulation. For example, the solubility of S in GeTe is very low, primarily forming precipitates, which have little influence on the charge carrier concentration and electronic energy band structure and thus barely enhance the power factor[62,63]. As a result, only a small ZT enhancement can be achieved due to the interface phonon scattering. Note that the phonon mean free path in GeTe is quite small[64]. The contribution of interface phonon scattering to a reduced thermal conductivity is very limited. In contrast, introducing point defects is more effective in impeding phonon propagation in GeTe. Therefore, it would be desirable to enhance the solubility of otherwise insoluble elements in GeTe and to study the effect of improved solubility on the charge and heat transport properties.

In this work, we have demonstrated that the chemical bonding mechanism can be applied to tailor the solubility of dopants and improve the thermoelectric performance. We prepared a series of GeTe-based samples such as $(GeTe)_{1-2x}(GeSe)_x(GeS)_x$, $(GeTe)_{1-2x}(SnSe)_x(SnS)_x$, and $(GeTe)_{1-2x}(PbSe)_x(PbS)_x$ to explore the solubility of dopants with different chemical bonding mechanisms and their concomitant thermoelectric properties. We observed sulfur-rich precipitates in the former two series of compounds when x is ≥1%. In striking contrast, no precipitates were observed in the last series of compounds even when x is greater than 5%. Consequently, a high ZT of 2.2 at 773 K was obtained in the solid solution of $(Ge_{0.84}Sb_{0.06}Te_{0.9})(PbSe)_{0.05}(PbS)_{0.05}$. The mechanisms underpinning the distinctive solubility behavior and the

improvement of thermoelectric properties are thoroughly investigated. We prove that employing the same metavalent bonding for the host and dopant is crucial for securing a high miscibility. Otherwise, phase separation is difficult to avoid. Accordingly, we propose a general doping strategy to tailor microstructures by understanding the chemical bonding mechanism of dopants and the host. This is of great significance to the design of functional materials beyond thermoelectrics.

## Results
### Mechanism of improved solubility
To determine the solubility of Sulfur in GeTe, we first prepared S-doped GeTe samples. The powder X-ray diffraction (XRD) patterns show the presence of S-rich second phases in the $GeTe_{0.98}S_{0.02}$ sample (Supplementary Fig. S1). The energy-dispersive spectroscopy (EDS) results also reveal S-rich second phases in $GeTe_{0.98}S_{0.02}$ and even in $GeTe_{0.99}S_{0.01}$ samples (Supplementary Figs. S2 and S3). This indicates that the solubility of S in GeTe is lower than 1%. Yet, Samanta et al.[63] and Acharyya et al.[62] reported a solubility of S higher than 1% in $(GeTe)_{1-2x}(GeSe)_x(GeS)_x$ and $(GeTe)_{1-2x}(SnSe)_x(SnS)_x$ according to their XRD results. This could imply that the increased configurational entropy can enhance the solubility limit of dopants compared to the singly doped $GeTe_{1-x}S_x$. This explanation has been discussed in other GeTe-based high-entropy alloys[56]. We also prepared a series of $(GeTe)_{1-2x}(GeSe)_x(GeS)_x$ and $(GeTe)_{1-2x}(SnSe)_x(SnS)_x$ samples to investigate the solubility behavior of sulfur. The XRD data (Fig. 1a and Supplementary Fig. S4) show the formation of S-rich second phases at a content of x > 1%. A close investigation of the microstructures using EDS (Fig. 1b and Supplementary Fig. S5 to S10) further confirms the low S solubility of less than 1% in these $(GeTe)_{1-2x}(GeSe)_x(GeS)_x$ and $(GeTe)_{1-2x}(SnSe)_x(SnS)_x$ samples. This implies that the increased configurational entropy does not necessarily enhance the solubility of S in GeTe. In contrast, we observed no impurity phases in the XRD patterns of $(GeTe)_{1-2x}(PbSe)_x(PbS)_x$ samples up to x = 7.5% (Fig. 1a) and in the EDS mapping of the sample x = 5% (Fig. 1c and Supplementary Fig. S11). Yet, we still observed a small fraction of S-rich precipitates in the sample $(GeTe)_{0.85}(PbSe)_{0.075}(PbS)_{0.075}$ by the more sensitive EDS. This indicates that the solubility of S in GeTe is enhanced to above 5% but below 7.5% by alloying with PbSe and PbS (Supplementary Fig. S12). Note that these series of $(GeTe)_{1-2x}(SnSe)_x(SnS)_x$ and $(GeTe)_{1-2x}(PbSe)_x(PbS)_x$ samples have the same nominal configurational entropy at the same x content. Yet, the solubility behavior of the dopants is distinctively different.

We then performed atom probe tomography (APT) to determine the chemical composition of the matrix and precipitates with very high chemical sensitivity at the level of ppm and spatial resolution down to the near-atomic scale[65–67]. Fig. 2a shows the distribution of S and Te in the sample $(GeTe)_{0.9}(SnSe)_{0.05}(SnS)_{0.05}$, while other elements are omitted for clarity. The top-right part shows a much higher number density of S atoms as depicted by red point clouds, indicating the presence of S-rich precipitates. The interface between the matrix and the precipitate is highlighted by an iso-composition surface of 2 at% S. The 3D-composition volume rendering of S (Fig. 2b) further indicates the striking contrast in the composition of S between the GeTe matrix and the S-rich precipitate. The proximity histogram using the 2 at% S iso-surface (Fig. 2c) shows that the S and Se composition in the precipitate can reach 15 at% and 10 at%, respectively. In contrast, only about 0.8 at% S and 2 at% Se are dissolved in the GeTe matrix, while the compositions of other elements are in line with their stoichiometries. Note that the analyzed volume of the precipitate is in the very vicinity of the GeTe matrix (about 5 nm). Thus, due to interfacial diffusion, we expect the measured composition to deviate from the thermodynamically stable composition of the precipitate. Therefore, we also prepared APT specimens including only individual precipitates to determine their chemical composition. Supplementary Fig. S13a, b show a homogeneous but higher content of S in the second phase of

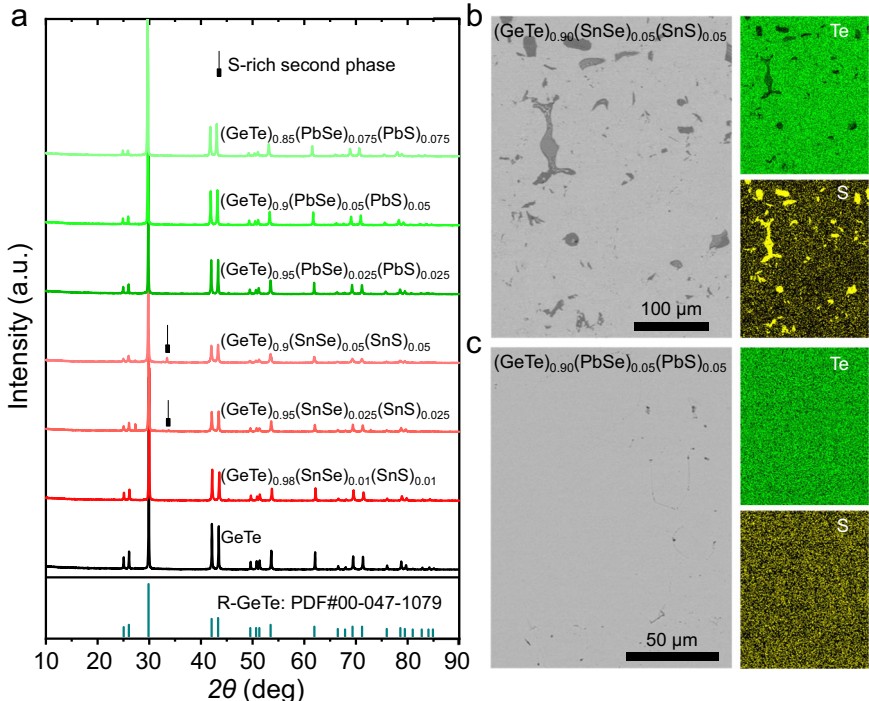

**Fig. 1 | Phase structure and microstructure. a** Room-temperature powder XRD patterns of $(GeTe)_{1-2x}(SnSe)_x(SnS)_x$, and $(GeTe)_{1-2x}(PbSe)_x(PbS)_x$; BSE images and EDS mapping for Te and S of (**b**) $(GeTe)_{0.9}(SnSe)_{0.05}(SnS)_{0.05}$ and (**c**) $(GeTe)_{0.9}(SnSe)_{0.05}(SnS)_{0.05}$.

the sample $(GeTe)_{0.9}(SnSe)_{0.05}(SnS)_{0.05}$. The corresponding 1D composition profile in Supplementary Fig. S13c demonstrates that the second phase is GeS-based with a high fraction of Se (10 at%) but a small fraction of Sn and Te (~3 at%). Very similar phenomena regarding the distribution and content of dopants in the matrix and second phase are also observed in the sample $(GeTe)_{0.9}(GeSe)_{0.05}(GeS)_{0.05}$, as presented in Supplementary Fig. S14.

Except for the high chemical sensitivity and spatial resolution, the APT technique can capture information on chemical bonding[68–72]. In the laser-assisted field evaporation mode, either a single ion or multiple ions evaporate, which is called a single event or multiple events, respectively. The ratio of the multiple events to the total number of events is named the "probability of multiple events (PME)"[73]. Fig. 2d shows the PME map of $(GeTe)_{0.9}(SnSe)_{0.05}(SnS)_{0.05}$. A high PME value (> 70%) is observed in the matrix. On the contrary, the S-rich second phase only shows a much lower PME value of <20%. Very similar phenomena regarding the PME in the matrix and second phase are also observed in the sample $(GeTe)_{0.9}(GeSe)_{0.05}(GeS)_{0.05}$, as displayed in Supplementary Fig. S14. It has been demonstrated that a high PME value (> 60%) is characteristic and a hallmark of metavalent bonding (MVB)[68,74,75]. Moreover, these metavalently bonded solids embrace a unique portfolio of properties such as moderate electrical conductivity, a large effective coordination number that violates the "8-N" rule, a large optical dielectric constant, a high Born effective charge, a high mode-specific Grüneisen parameter, and a small band gap[76–79]. This unconventional combination of properties and abnormal bond-rupture behavior differentiates MVB from well-known metallic, covalent, and ionic bonding[76,77].

Different from the phase separation observed in $(GeTe)_{0.9}(SnSe)_{0.05}(SnS)_{0.05}$ and $(GeTe)_{0.9}(GeSe)_{0.05}(GeS)_{0.05}$ samples, the $(GeTe)_{0.9}(PbSe)_{0.05}(PbS)_{0.05}$ sample shows a single-phase solid solution across multiple scales. Supplementary Fig. S15a shows a typical domain structure of GeTe-based materials generated during the phase transition from the cubic structure to the rhombohedral structure. The corresponding EDS elemental mappings (Supplementary Fig. S15b) display a homogeneous distribution of all elements. Geometric phase

analysis (GPA) shows a large strain in the domain boundary, which can effectively scatter heat-carrying phonons with low and intermediate frequencies[80] (Supplementary Fig. S15c and S15d). The 3D distribution of S (Fig. 2e) and the corresponding volume rendering (Fig. 2f) as well as the composition profile (Fig. 2g) of elements obtained by APT all confirm the homogeneous and accurate composition in the $(GeTe)_{0.9}(PbSe)_{0.05}(PbS)_{0.05}$ sample. Moreover, this sample maintains a high PME value (>70%) as observed in the GeTe matrix[41](Fig. 2h), indicating that the high solubility of these dopants does not change the bonding mechanism. These results reveal that the improvement of the solubility of dopants is independent of the increased configurational entropy because the two samples $(GeTe)_{0.9}(SnSe)_{0.05}(SnS)_{0.05}$ and $(GeTe)_{0.9}(PbSe)_{0.05}(PbS)_{0.05}$ have the same nominal configurational entropy. This calls for additional mechanisms to underpin the improved solubility of S in GeTe.

By carefully comparing the composition and PME maps in Fig. 2, it appears that the phase separation is accompanied by a change in the chemical bonding mechanism. Due to the development of quantum-mechanical calculation tools, the chemical bonding of solid-state materials can be quantified based on the number of electrons shared (ES) between adjacent atoms and electrons transferred (ET) from one atom to its neighbors[81], as shown in Fig. 3a. This 2D map is spanned by ET normalized by the oxidation state of elements and ES. It is noteworthy that different chemical bonding mechanisms such as metallic, covalent, and ionic bonds can be separated by ES and ET in the map. Specifically, metavalently bonded solids prevail in a well-defined region characterized by a small ET value and an ES value close to 1, as depicted in green in the map[50,69,81]. Hence, MVB is characterized by sharing about one electron, i.e., half an electron pair. From this chemical bonding map, we can find that GeTe, PbSe and PbS employ MVB, whereas GeSe, GeS, SnSe and SnS utilize covalent bonding. This map offers an explanation for the microstructures observed by the different characterization techniques discussed above. Alloying metavalently bonded GeTe with PbSe and PbS forms a single-phase solid solution over a large composition range. In contrast, the covalently bonded GeSe, GeS, SnSe and SnS compounds are hardly dissolvable in the

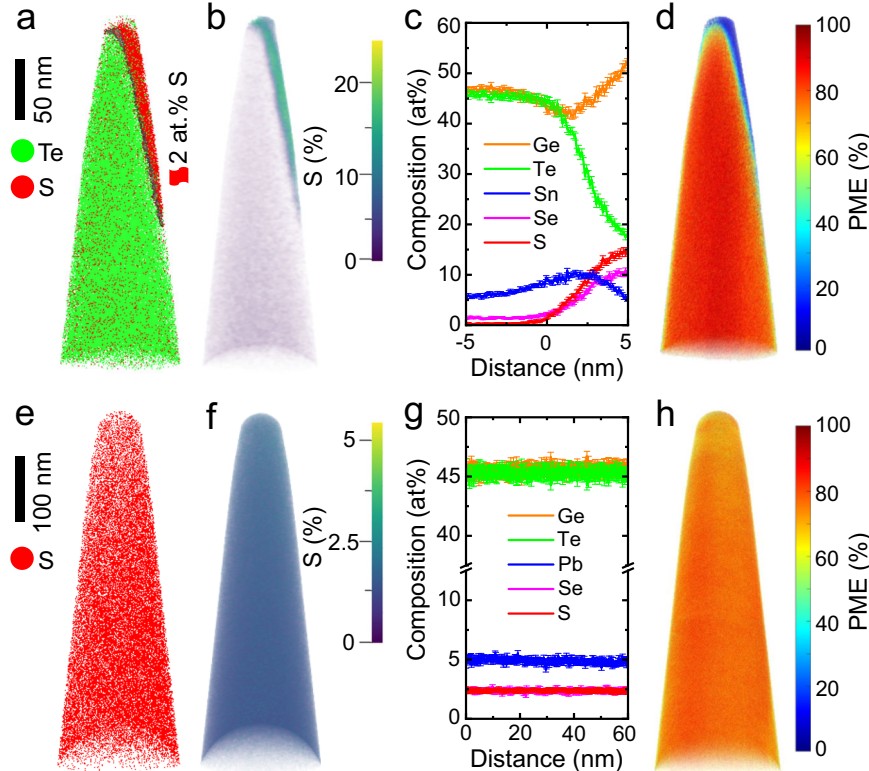

**Fig. 2 | Microstructures of $(GeTe)_{0.9}(SnSe)_{0.05}(SnS)_{0.05}$ and $(GeTe)_{0.9}(PbSe)_{0.05}(PbS)_{0.05}$ samples. a** 3D reconstruction showing the distribution of Te and S elements and the interface between second phase and matrix highlighted by an iso-composition surface of 2 at% S, (**b**) volume rendering showing the composition of S in 3D space, (**c**) composition proximity histogram across the interface between second phase and matrix, (**d**) 3D PME map indicating the distinct bond-breaking behaviors for second phase and matrix of $(GeTe)_{0.9}(SnSe)_{0.05}(SnS)_{0.05}$; (**e**) 3D reconstruction showing the distribution of S element, (**f**) volume rendering showing the composition of S in 3D space, (**g**) composition profile of elements taken from a cuboid region of interest along the vertical direction, (**h**) 3D PME map of $(GeTe)_{0.9}(PbSe)_{0.05}(PbS)_{0.05}$.

metavalently bonded GeTe, as shown schematically in Fig. 3b. Note that the covalently bonded selenides show higher solubility than that of covalently bonded sulfides in GeTe due to the smaller differences in atomic radii and electronegativity between Te and Se. Nevertheless, the solubility of PbSe in GeTe is still larger than that of SnSe and GeSe in GeTe. Moreover, the improved solubility of S in GeTe is primarily attributed to the same MVB mechanism between GeTe and PbS as proven above. One might argue that the different miscibility behavior is due to the different crystal structures. Indeed, the different crystal structures of these chalcogenides are determined by their different chemical bonding mechanisms. Metavalent bonding is characterized by a half-filled σ-bond constructed by the overlap of p-orbitals[76]. Due to the orthogonal alignment of p-orbitals, an ideal MVB solid should utilize a cubic structure. Yet, this configuration is energetically unstable, which spontaneously creates Peierls distortion to lower the energy of the system[82]. Thus, rhombohedral GeTe is more stable under ambient conditions. Yet, the cubic structure can also be stabilized by increasing the charge transfer, which explains the rock-salt structure of PbSe and PbS. In striking contrast, the p-orbital overlap for GeS and SnS is much smaller than that in GeTe, leading to a significantly larger degree of Peierls distortion. This results in a structural phase transition from rhombohedral to orthorhombic associated with a chemical bonding transition from metavalent to covalent. In this regard, the low miscibility between materials with different crystal structures can also be ascribed to the different bonding mechanisms. In contrast, even though the crystal structures between rhombohedral GeTe and cubic PbS are different, they can still form a solid solution due to their same metavalent bonding mechanism.

To check the general applicability of the atomic doping strategy in other MVB materials, we selected MVB SnTe as another example. The metavalent bonding nature of SnTe has been proven by its unique property fingerprints and high PME value measured by APT in previous studies[67,72,77]. The solubility of S in the SnTe is lower than 1%, as shown in Supplementary Fig. S16 and Fig. S17. However, using MVB PbSe and PbS alloyed into SnTe, the solubility of S in the matrix is higher than 2.5% (Supplementary Fig. S18 and Fig. S19). In addition, our proposed approach can also explain the high solubility of $Sb_2Te_3$ or $Bi_2Se_3$ in $Bi_2Te_3$ since they all employ MVB[49]. On the contrary, the covalently bonded $Bi_2S_3$ and $Sb_2Se_3$ are immiscible with the metavalently bonded $Bi_2Te_3$[23,50]. Besides, alloying ionic bonding materials with MVB materials also prefers to form precipitates, such as SrTe/CaTe/BaTe precipitates (ionic bonding) observed in PbTe (MVB) even though both the precipitates and matrix utilize a cubic structure[20,83]. Therefore, according to our experiments and previous reports, our proposed selection strategy of dopants for MVB materials based on the chemical bonding mechanism has general applicability. As a consequence, chemical bonding can be applied as an indicator to tailor the solubility of foreign atoms to tune the properties of functional materials such as thermoelectric materials, phase-change materials, and optoelectronic materials.

## Thermoelectric transport properties

The different solubility behavior of S and associated microstructures result in different thermoelectric transport properties. Fig. 4a compares the carrier concentration for $(GeTe)_{1-2x}(GeSe)_x(GeS)_x$, $(GeTe)_{1-2x}(SnSe)_x(SnS)_x$ and $(GeTe)_{1-2x}(PbSe)_x(PbS)_x$ samples as a function of $x$. In the case of Pb-included samples, the carrier

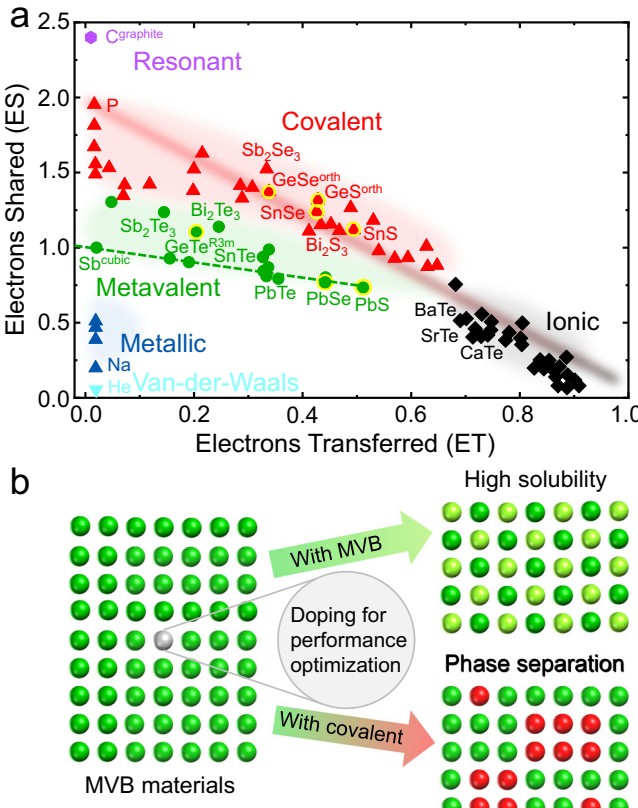

**a**

**b**

**Fig. 3 | Chemical bonding classification and schematic illustration of the design of microstructures by alloying materials with different chemical bonding mechanisms. a** 2D map classifying chemical bonding in solids. The map is spanned by the electrons transferred (ET) normalized by the oxidation state and the electrons shared between adjacent atoms, and the different colors are employed to describe different chemical bonding mechanisms. Data are adapted from refs. 50,69,81. (**b**) Schematic illustration of the way to obtain a solid solution or phase separation by alloying materials with different bonding mechanisms.

concentration decreases with increasing the content of PbSe and PbS. Supplementary Fig. S20 shows the backscattering images of $(GeTe)_{1-2x}(PbSe)_x(PbS)_x$ samples. A secondary Ge phase exists in all samples, which is due to the low formation energy of Ge vacancies originating from the weak chemical bonds and the large difference between cation and anion sizes in GeTe[84,85]. Since alloying GeTe with PbSe and PbS can reduce the difference between cation and anion sizes, the formation energy of Ge vacancies is enhanced[61]. Consequently, the carrier concentration and the content of Ge second phase in $(GeTe)_{1-2x}(PbSe)_x(PbS)_x$ decrease with $x$ increasing, as shown in Fig. 4a and Supplementary Fig. S21. In contrast, the carrier concentrations of $(GeTe)_{1-2x}(GeSe)_x(GeS)_x$ and $(GeTe)_{1-2x}(SnSe)_x(SnS)_x$ increase upon increasing $x$ (Fig. 4a). This could be due to the formation of GeS-rich secondary phases, which consume Ge from the GeTe matrix, as can be indicated by the relatively higher content of Ge in the precipitates than that in the matrix determined by APT. Thus, a higher content of cation vacancies and thus hole carrier concentration is generated. The charge carrier mobility often shows the opposite trend as to the carrier concentration, which is also observed in the cases of $(GeTe)_{1-2x}(GeSe)_x(GeS)_x$ and $(GeTe)_{1-2x}(SnSe)_x(SnS)_x$, as shown in Fig. 4b. Yet, the carrier mobility of the $(GeTe)_{1-2x}(PbSe)_x(PbS)_x$ samples decrease with increasing $x$ even though the carrier concentration is also decreased. This is mainly because of the increased content of substitutional point defects due to the formation of a solid solution.

Although the carrier mobility is relatively higher in $(GeTe)_{0.9}(PbSe)_{0.05}(PbS)_{0.05}$, its significantly reduced carrier

concentration contributes to a lower $\sigma$ compared with that of GeTe, $(GeTe)_{0.9}(GeSe)_{0.05}(GeS)_{0.05}$, and $(GeTe)_{0.9}(SnSe)_{0.05}(SnS)_{0.05}$ (Fig. 4c). Because of the inverse relationship between carrier concentration and $\alpha$, the $\alpha$ of $(GeTe)_{0.9}(PbSe)_{0.05}(PbS)_{0.05}$ is the highest among compounds GeTe, $(GeTe)_{0.9}(PbSe)_{0.05}(PbS)_{0.05}$, $(GeTe)_{0.9}(GeSe)_{0.05}$ $(GeS)_{0.05}$, and $(GeTe)_{0.9}(SnSe)_{0.05}(SnS)_{0.05}$ (Fig. 4d). Moreover, the enhancement of $\alpha$ is also related to the increase of density-of-states effective mass ($m^*$) (Supplementary Fig. S22), which is due to that PbSe and PbS alloying increases the interaxial angles and facilitates band convergence, as demonstrated in Supplementary Fig. S23. The reduced $\sigma$ generates low $\kappa_e$ and $\kappa_{tot}$ (Supplementary Fig. S24). Moreover, the $\kappa_L$ of $(GeTe)_{0.9}(PbSe)_{0.05}(PbS)_{0.05}$ is lower than other materials (Fig. 4e), which can be ascribed to three factors. First, the alloying effect introduces large mass and strain fluctuations between the host and doping atoms. Second, MVB materials have strong lattice vibration anharmonicity, which increases the Umklapp phonon scattering[86]. Third, alloying GeTe with PbSe and PbS induces lattice softening, leading to reduced sound velocity, as experimentally verified in Supplementary Fig. S25. To prove the strengthened anharmonicity, Grüneisen parameters ($\gamma$) were calculated based on the sound velocities measured. Computational details and results are shown in the Supplementary Text and Table S1, respectively. The $\gamma$ increases from 1.43 for GeTe to 1.56 for $(GeTe)_{0.85}(PbSe)_{0.075}(PbS)_{0.075}$, indicating the increase of lattice vibration anharmonicity. Fig. 4f shows the temperature-dependent $ZT$ for GeTe, $(GeTe)_{0.9}(PbSe)_{0.05}(PbS)_{0.05}$, $(GeTe)_{0.9}(GeSe)_{0.05}(GeS)_{0.05}$, and $(GeTe)_{0.9}(SnSe)_{0.05}(SnS)_{0.05}$ samples. The $ZT$ of $(GeTe)_{0.9}(PbSe)_{0.05}(PbS)_{0.05}$ is significantly larger compared to other compounds due to the optimization of carrier concentration and the reduction of $\kappa_L$, while the $ZT$ of $(GeTe)_{0.9}(GeSe)_{0.05}(GeS)_{0.05}$, and $(GeTe)_{0.9}(SnSe)_{0.05}(SnS)_{0.05}$ is comparable or even lower than that of pristine GeTe. This indicates that alloying GeTe with GeSe and GeS or SnSe and SnS hardly improves the performance due to the very small solubility limit of dopants. Meanwhile, the GeS-rich second phase does not improve the thermoelectric performance of GeTe as well. On one hand, the more resistive second phase can only improve the overall thermoelectric performance if its size is comparable to the mean free path of phonons, which reduces the lattice thermal conductivity. Yet, the GeS-rich second phases observed in this work are on micrometer scales, indicating weak effects on scattering phonons in GeTe with an intrinsically small phonon mean-free path. On the other hand, the thermoelectric performance of GeS with covalent bonding is lower than GeTe with metavalent bonding[86]. The composition of GeS into GeTe can hardly improve the overall thermoelectric performance. The average $ZT_{ave}$ between 300 and 773 K of $(GeTe)_{0.9}(PbSe)_{0.05}(PbS)_{0.05}$ is 2.1 times that of pure GeTe, 1.8 times that of $(GeTe)_{0.9}(GeSe)_{0.05}(GeS)_{0.05}$, and 3.0 times that of $(GeTe)_{0.9}(SnSe)_{0.05}(SnS)_{0.05}$, as shown in the inset of Fig. 4f. More systematic analyses of the thermoelectric properties of $(GeTe)_{1-2x}(PbSe)_x(PbS)_x$ samples with $x$ ranges from 0 to 0.075 can be found in the Supporting Information (Supplementary Fig. S21–S28).

Considering the carrier concentration of $(GeTe)_{0.9}(PbSe)_{0.05}$ $(PbS)_{0.05}$ ($\sim5 \times 10^{20}\,cm^{-3}$) exceeds the optimal carrier concentration range for GeTe ($1-2 \times 10^{20}\,cm^{-3}$) [87], Sb donor doping is further introduced to meticulously tune the carrier concentration. Note that the high solubility of Sb in GeTe is also partly due to the same MVB mechanism for both Sb and GeTe[41]. Supplementary Fig. S29 displays the powder XRD patterns of $(Ge_{0.9-y}Sb_yTe_{0.9})(PbSe)_{0.05}(PbS)_{0.05}$ samples as well as the lattice parameters and inter-axial angles obtained by Rietveld refinement. The main phase is R-GeTe, and a small amount of a secondary Ge phase appears, which is a usual phenomenon in Sb-doped GeTe-based materials[57,88]. The lattice parameters increase with increasing the doping content, indicating that Sb has been substituted to the lattice sites.

Figure 5a shows that the carrier concentration and carrier mobility of $(Ge_{0.9-y}Sb_yTe_{0.9})(PbSe)_{0.05}(PbS)_{0.05}$ samples are reduced

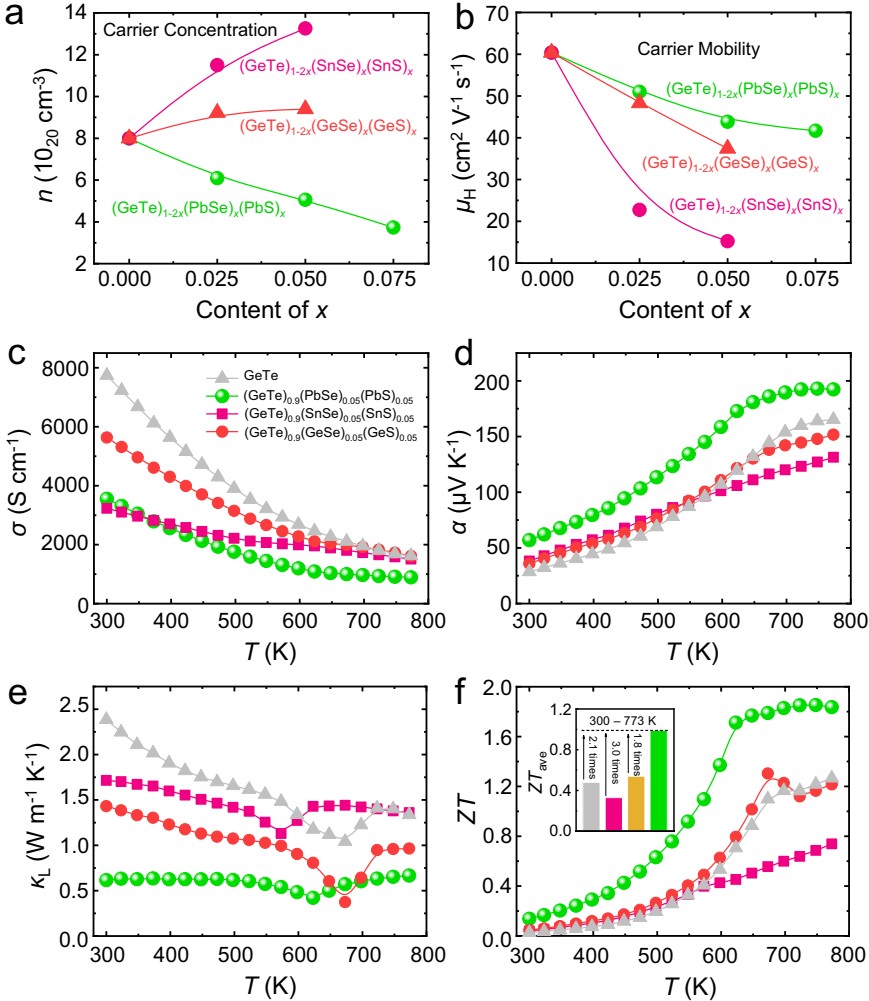

**Fig. 4 | Thermoelectric transport properties. a** Carrier concentration, (**b**) carrier mobility of $(GeTe)_{1-2x}(GeSe)_x(GeS)_x$, $(GeTe)_{1-2x}(SnSe)_x(SnS)_x$, and $(GeTe)_{1-2x}(PbSe)_x(PbS)_x$; (**c**) Electrical conductivity, (**d**) Seebeck coefficient, (**e**) lattice thermal conductivity, (**f**) $ZT$ value of GeTe, $(GeTe)_{0.9}(PbSe)_{0.05}(PbS)_{0.05}$, $(GeTe)_{0.9}(GeSe)_{0.05}(GeS)_{0.05}$, and $(GeTe)_{0.9}(SnSe)_{0.05}(SnS)_{0.05}$. The inset of (**f**) is $ZT_{zve}$ of GeTe, $(GeTe)_{0.9}(PbSe)_{0.05}(PbS)_{0.05}$, $(GeTe)_{0.9}(GeSe)_{0.05}(GeS)_{0.05}$, and $(GeTe)_{0.9}(SnSe)_{0.05}(SnS)_{0.05}$.

with doping Sb. The decrease in $n_H$ is due to the one more valence electron of Sb than Ge, behaving as an electron donor. The slight reduction of carrier mobility with increasing the content of Sb is ascribed to the compromise between the increased impurity scattering and the weakened carrier scattering. On one hand, the increased content of Sb dopants could enhance the alloy scattering of electrons and thus reduce the carrier mobility. On the other hand, the reduction of carrier concentration of Sb doping weakens the scattering of electrons and then increases the carrier mobility. Owing to the reduction of carrier concentration and mobility, the $\sigma$ of $(Ge_{0.9-y}Sb_yTe_{0.9})$ $(PbSe)_{0.05}(PbS)_{0.05}$ samples is significantly decreased with increasing the content of Sb (Fig. 5b). The $\alpha$ is increased due to the reduction of carrier concentration (Fig. 5c). The $\kappa_{tot}$ (Supplementary Fig. S30) after Sb doping reduces, which is mainly contributed by the decrease of $\kappa_e$ rooted in the reduction of $\sigma$ according to the Wiedemann-Franz law. In addition, the variation of $\kappa_L$ (Fig. 5d) and sound velocity (Supplementary Fig. S31) after Sb doping is small, which implies no significant changes in the phonon-point defect scattering strength and the chemical bonding mechanism. In the end, due to the optimization of carrier concentration, the $ZT$ in the whole measured temperature range, especially at the low and intermediate temperatures, is improved after Sb doping. A peak $ZT$ of 2.2 at 773 K is realized in $(Ge_{0.84}Sb_{0.06}Te_{0.9})(PbSe)_{0.05}(PbS)_{0.05}$ (Fig. 5e). As demonstrated in

Fig. 5f, the high $ZT$ comes from the enhanced solubility of dopants caused by the chemical bonding mechanism.

## Discussion

In this study, we propose a selection strategy of dopants for MVB materials to control microstructures by comparing the chemical bonding mechanisms between the host and dopants. We find that alloying MVB solids can largely improve the solid solubility of dopants. In contrast, phase separation occurs if MVB materials are mixed with covalently bonded or ionically bonded solids. This is demonstrated in MVB GeTe thermoelectrics. We respectively use the covalently bonded GeSe and GeS, SnSe and SnS, as well as metavalently bonded PbSe and PbS to alloy with GeTe. We find that the solubility of S element in $(GeTe)_{1-2x}(GeSe)_x(GeS)_x$ and $(GeTe)_{1-2x}(SnSe)_x(SnS)_x$ is less than 1%, while the solubility of S is distinctly enhanced (>5%) in $(GeTe)_{1-2x}(PbSe)_x(PbS)_x$. The electrical transport properties are optimized and the lattice thermal conductivity is reduced due to the enhanced solubility. In conjunction with the optimized carrier concentration by Sb donor doping, a high $ZT$ of 2.2 at 773 K is achieved in $(Ge_{0.84}Sb_{0.06}Te_{0.9})(PbSe)_{0.05}(PbS)_{0.05}$. Moreover, this strategy has been cross-confirmed in other compounds such as MVB SnTe (in this work), MVB $Bi_2Te_3$ (in literature), and MVB PbTe (in literature). This work develops a doping strategy to either form a solid solution or

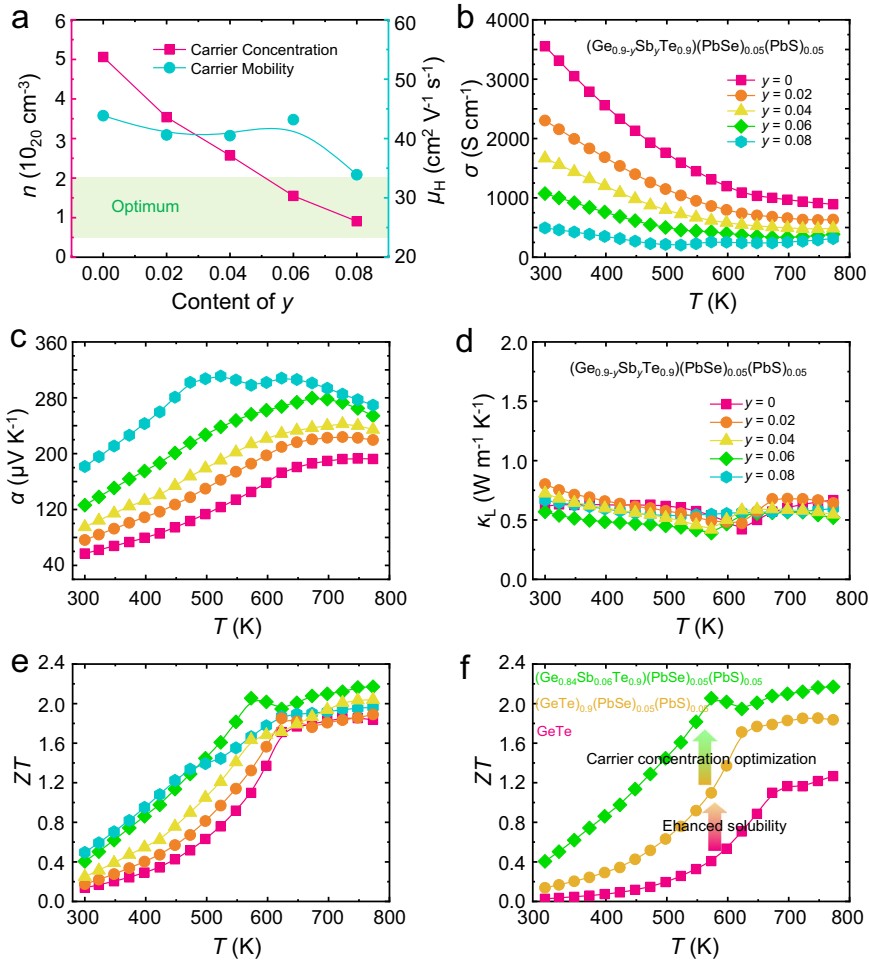

**Fig. 5 | Thermoelectric transport properties. a** Carrier concentration and carrier mobility of $(Ge_{0.9-y}Sb_yTe_{0.9})(PbSe)_{0.05}(PbS)_{0.05}$; (**b**) Electrical conductivity, (**c**) Seebeck coefficient, (**d**) lattice thermal conductivity, (**e**) ZT value of $(Ge_{0.9-y}Sb_yTe_{0.9})(PbSe)_{0.05}(PbS)_{0.05}$; (**f**) ZT value of pure and optimized GeTe-based materials.

phase separation by understanding the chemical bonding mechanisms, providing insights into tailoring the desired properties upon microstructural design for MVB materials.

## Methods
### Sample preparation
High-purity elements (Ge, Te, Pb, Se, S, Sb, and Sn) were weighed in a glove box filled with argon based on the nominal compositions. The mixtures were encapsulated within silica tubes under stringent vacuum conditions. These ampoules underwent a heating process at 1273 K for 12 h, followed by a gradual cooling to 873 K for 4 h, and then maintained at this temperature for 3 days. Afterwards, the solidified ingots were pulverized utilizing an agate mortar and consolidated via Spark Plasma Sintering (SPS) at 823 K for 5 min by applying a pressure of 60 MPa.

### Sample characterization
The phase structure was analyzed at room temperature using powder X-ray diffraction (X'Pert PRO-PANalytical, Netherlands), with the lattice parameters ($a$) and interaxial angles ($\beta$) being precisely determined through the application of the Rietveld refinement method. Furthermore, the microstructural features and elemental distribution were examined by scanning electron microscopy (ZEISS, Germany) and transmission electron microscopy (Talos f200x, United States), respectively. For APT measurements, the needle-shaped specimens were prepared by SEM-FIB dual beam focused ion beam (Helios 650, FEI) using the standard "lift-out" method. APT measurements were conducted on LEAP 4000X Si (CAMECA) by employing a UV laser (wavelength = 355 nm) pulse with a laser pulse energy of 10 pJ, a pulse repetition rate of 200 KHz, a specimen base temperature of 40 K, a detection rate of 1.0% on average, and an ion flight path of 160 mm. APT data reconstruction was processed with the commercial software IVAS 3.8. The electrical transport characteristics, encompassing the Seebeck coefficient and electrical conductivity, were concurrently measured using a commercial instrument (ZEM-3, Japan). The thermal conductivity ($\kappa$) was derived from the equation $\kappa = \lambda \cdot C_p \cdot d$, where $\lambda$ stands for thermal diffusivity, $C_p$ is the specific heat, and $d$ represents the density of the material. The thermal diffusivity ($\lambda$) was recorded employing the laser flash diffusivity method (LFA 457, Germany), while the specific heat ($C_p$) was estimated using the Dulong-Petit law, and the density ($d$) was determined through the Archimedes method. Furthermore, the sound velocity was assessed utilizing the ultrasonic reflection method (UMS-100, France). The Hall coefficient ($R_H$) was measured via the van der Pauw technique under a reversible magnetic field of 1.5 T. Subsequently, the Hall carrier concentration ($n$) and mobility ($\mu_H$) were calculated using the formulas $n = 1/(eR_H)$ and $\mu_H = R_H\sigma$, respectively, where $e$ denotes the electron charge.

### Density functional calculation
DFT calculation was employed using the projector augmented wave (PAW) method in the Vienna Ab-initio Simulation Package (VASP), and the exchange-correlation functional of Perdew-Burke-Ernzerhof (PBE) generalized gradient approximation (GGA) was implemented. We constructed a $3 \times 3 \times 3$ supercell and a k-point mesh of $4 \times 4 \times 4$ was

used for calculating band structures. The kinetic energy cutoff of 450 eV is used to truncate the plane wave basis and atomic forces were smaller than 0.001 eV Å$^{-1}$ during structural relaxation.

## Reporting summary

Further information on research design is available in the Nature Portfolio Reporting Summary linked to this article.

## Data availability

All data necessary to understand and assess this manuscript are shown in the main text and the Supporting Information. The data that support the findings of this study are available from the corresponding author upon reasonable request.

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

## Acknowledgements

This work was supported by the National Natural Science Foundation of China (Nos. 52271206, 52130106, 52101247, and 523B2020), the Natural Science Foundation of Sichuan Province (No. 2024NSFSC0992), and the Fundamental Research Funds for the Central Universities (HIT.DZJJ.2024003).

## Author contributions

M.L., Y.Yu., and J.S. developed the concept and designed the experiments. M.L. and M.G. carried out the theoretical calculations. M.L., M.G., H.L., Y.L., Y.Z., F.G., Y.Yang., K.Y., X.D., Z.L., and W.C. performed the experiments and measurements. M.W. Y.Yu. and J.S. supervised the whole project. M.L. and Y.Yu. wrote the draft. All authors discussed the results and commented on the manuscript.

## Funding

## Competing interests

The authors declare no competing interests.
