## [Peer Review File · Nature Communications]

Doping Strategy in Metavalently Bonded Materials for Advancing Thermoelectric PerformanceREVIEWER COMMENTS

Reviewer #1 (Remarks to the Author):

This manuscript proposed that chemical bonding mechanism can be applied to determine the solubility of dopants in metavalent bonding materials and proved this theory by preparing and contrasting a series of GeTe-based samples such as $(\text{GeTe})_{1-2x}(\text{GeSe})_x(\text{GeS})_x$, $(\text{GeTe})_{1-2x}(\text{SnSe})_x(\text{SnS})_x$, and $(\text{GeTe})_{1-2x}(\text{PbSe})_x(\text{PbS})_x$, which is innovative and interesting, and is also checked in other metavalent bonding materials such as SnTe, PbTe, and Bi_2Te_3 . In addition, due to the realized high solubility of PbSe, PbS, and Sb in GeTe, the thermoelectric transport properties were optimized and a high ZT of 2.2 was realized. This result is beneficial for the optimization of other thermoelectric materials through chemical bonding design. Thus, I recommend the paper to be published in Nature Communications after addressing the following issues.

1. There is not ET and ES of SnTe in Fig. 3a. I don't know if SnTe is metavalent bonding material. Please add the relevant data.
2. The carrier concentration in $(\text{GeTe})_{1-2x}(\text{PbSe})_x(\text{PbS})_x$ decreases with x increasing. The author thinks that it is related to the strengthened chemical bond. But this conclusion is contrast with lattice softening.
3. The lattice thermal conductivity reduction of $(\text{GeTe})_{1-2x}(\text{PbSe})_x(\text{PbS})_x$ is not only attributed to enhanced point defect scattering but also attributed to the strong anharmonicity. However, there are no direct evidence.
4. In this manuscript, the symbols of both Seebeck coefficient and interaxial angles use α . In addition, the symbol of the Seebeck coefficient in Supplementary Note is S. Please check for other errors.
5. In previous papers (Joule, 2018, 2, 976-987; Cell Reports Physical Science, 2022, 3, 101009), increasing interaxial angles is proved to be helpful in converging energy band in GeTe. In this manuscript, interaxial angles also increase with PbSe, PbS and Sb alloying/doping. Is there band convergence?

Reviewer #2 (Remarks to the Author):

This manuscript discusses an important problem in thermoelectric materials and beyond, specifically the mechanisms of doping or alloying. The authors propose that compounds with metavalent bonding easily mix to form solid solutions. The idea has been tested in the GeTe-PbSe-PbS system, which forms a single-phase solid solution, in contrast to GeTe-GeSe-GeS and GeTe-SnSe-SnS system. This idea is intriguing, and I recommend it to be published after addressing the following questions.

- (1) For the difference between PbS alloyed GeTe and SnS/GeS alloyed ones, have the authors considered the different crystal structures of SnS and GeS? Typically, compounds with different crystal structures are harder to form a solid solution.
- (2) PbS is a metavalent-bonding compound, whereas GeS and SnS are not. Therefore, it can be expected that in the local environment of S in GeTe-PbS system, Pb should be present. Is it possible that S-Pb pairs could form in this system.
- (3) The increased carrier concentration in GeTe-GeSe-GeS system has been attributed to the formation of GeS-rich secondary phases. However, GeS is not expected to consume Ge from GeTe matrix as it is added in the form of GeS, not S.
- (4) The decrease in lattice thermal conductivity in the GeTe-PbSe-PbS system could be caused by the stronger anharmonicity, which can be further testified by the higher Gruneisen

constant. The author can calculate it through sound velocities.

Reviewer #3 (Remarks to the Author):

This work proposed a doping strategy to tailor microstructures for metavalently bonded materials by understanding the relationship between the solubility of dopants and the chemical bonding mechanism. The authors demonstrated that high solubility can be achieved by alloying metavalently bonded solids. On the contrary, secondary phases are formed by alloying materials employing different chemical bonding mechanisms. This conclusion has been well demonstrated in a series of GeTe-based alloys. The high solubility of PbSe and PbS in GeTe is attributed to the metavalent bonding mechanism of these compounds. The generality has also been confirmed in SnTe-based alloys and applied to explain similar phenomena in other systems reported such as Bi₂Te₃ and PbTe. As a result of the microstructure manipulation, the authors proved that a high solubility of PbSe and PbS in GeTe leads to an improved thermoelectric performance. This work broadens the understanding of microstructure and property manipulation by chemical doping. I am also very impressed by the beautiful data and figures. Therefore, I recommend the publication of this manuscript in Nature Communications. The following points should be addressed before the final publication.

1. In thermoelectrics, constructing a second phase is a common way to improve thermoelectric performance. Why does the second phase such as GeS not improve the thermoelectric properties of GeTe in this work?
2. The lattice thermal conductivity in the GeTe sample alloyed with PbSe and PbS is significantly reduced. Besides strong point defect scattering, the change of phonon dispersion can also lead to the reduction of thermal conductivity. What is the effect of Pb, Se, S doping or PbSe and PbS alloying on the phonon dispersion of GeTe?
3. Generally, doping-induced ionized impurity scattering will reduce carrier mobility. Why does Sb doping not significantly reduce carrier mobility?
4. There is a small diffraction peak in the 20-30° of XRD patterns (Fig. S1). What is that?
5. There are a large number of domain structures in (GeTe)_{0.9}(PbSe)_{0.05}(PbS)_{0.05} sample (Fig. S15). What effect do they have on thermal transport property?
6. As can be seen from Fig. S25, the m^* is increased with PbSe and PbS alloying, what are the effects of Pb, Se, and S doping on the band structure of GeTe?
7. More details of the synthesis, characterization, and calculation should be provided (in the SI).
8. Some newly reported papers about GeTe should be mentioned in the manuscript, e.g., Adv. Energy Mater. 2024, 14 (26), 2400340; J. Am. Chem. Soc. 2024, 146 (2), 1681; Adv. Funct. Mater. 2023, 33 (25), 2301750.

Reviewer #1 (Remarks to the Author):

This manuscript proposed that chemical bonding mechanism can be applied to determine the solubility of dopants in metavalent bonding materials and proved this theory by preparing and contrasting a series of GeTe-based samples such as $(\text{GeTe})_{1-2x}(\text{GeSe})_x(\text{GeS})_x$, $(\text{GeTe})_{1-2x}(\text{SnSe})_x(\text{SnS})_x$, and $(\text{GeTe})_{1-2x}(\text{PbSe})_x(\text{PbS})_x$, which is innovative and interesting, and is also checked in other metavalent bonding materials such as SnTe, PbTe, and Bi_2Te_3 . In addition, due to the realized high solubility of PbSe, PbS, and Sb in GeTe, the thermoelectric transport properties were optimized and a high ZT of 2.2 was realized. This result is beneficial for the optimization of other thermoelectric materials through chemical bonding design. Thus, I recommend the paper to be published in Nature Communications after addressing the following issues.

Response: Thanks for your positive evaluation of this work. We have revised the manuscript based on your comments. Please find the point-by-point responses below.

1. There is not ET and ES of SnTe in Fig. 3a. I don't know if SnTe is metavalent bonding material. Please add the relevant data.

Response: Thanks for your reminder. We have added the ET and ES of SnTe in Fig. 3a, as indicated by the purple mark in Fig. R1. SnTe is a typical metavalently bonded material, which can be determined by its unique property portfolio such as a large optical dielectric constant, a high Born effective charge, and strong anharmonicity of the transverse optical phonon mode [Adv. Mater., 2018, 30, 1803777]. Besides, SnTe shows an abnormal bond-breaking behavior in laser-assisted atom probe tomography measurements, exhibiting a PME (probability of multiple events) value of 75% [Adv. Funct. Mater. 2022, 32, 2209980]. All these properties in conjunction with the ES and ET values corroborate the metavalent bonding nature of SnTe.

Revision: We have indicated the position of SnTe in the ES-ET map in Fig. 3. We have also emphasized that SnTe is a metavalently bonded compound in the revised manuscript on page 8: “*The metavalent bonding nature of SnTe has been proven by its unique property fingerprints and high PME value measured by APT in previous*

studies.”.

Fig. R1. 2D map classifying chemical bonding in solids. The map is spanned by the electrons transferred (ET) normalized by the oxidation state and the electrons shared (ES) between adjacent atoms.

2. The carrier concentration in $(\text{GeTe})_{1-2x}(\text{PbSe})_x(\text{PbS})_x$ decreases with x increasing. The author thinks that it is related to the strengthened chemical bond. But this conclusion is contrast with lattice softening.

Response: This is a very insightful comment. By carefully investigating the formation mechanism of free charge carriers in GeTe, we have realized that the decreased hole concentration in $(\text{GeTe})_{1-2x}(\text{PbSe})_x(\text{PbS})_x$ with increasing x could be due to the increased formation energy of cation vacancies. Both the substitution of Ge by Pb and Te by Se/S increase the formation energy of cation vacancies [Adv. Sci., 2017, 4, 1700341]. Therefore, the carrier concentration in $(\text{GeTe})_{1-2x}(\text{PbSe})_x(\text{PbS})_x$ decreases with x increasing.

Revision: We have explained the decreased carrier concentration with increasing x for $(\text{GeTe})_{1-2x}(\text{PbSe})_x(\text{PbS})_x$ on Page 11: “A secondary Ge phase exists in all samples, which is due to the low formation energy of Ge vacancies originating from the weak chemical bonds and the large difference between cation and anion sizes in GeTe^{78, 79}. Since alloying GeTe with PbSe and PbS can reduce the difference between cation and

anion sizes, the formation energy of Ge vacancies is enhanced⁵⁷. Therefore, the carrier concentration and the content of Ge second phase in $(\text{GeTe})_{1-2x}(\text{PbSe})_x(\text{PbS})_x$ decrease with x increasing.”

3. The lattice thermal conductivity reduction of $(\text{GeTe})_{1-2x}(\text{PbSe})_x(\text{PbS})_x$ is not only attributed to enhanced point defect scattering but also attributed to the strong anharmonicity. However, there are no direct evidence.

Response: We have calculated the Grüneisen parameters (γ) (the larger the γ , the stronger the anharmonicity) based on the sound velocity measured. Corresponding computational details and results are shown in the Supplementary Text and Table S1, respectively. The γ increases from 1.43 for GeTe to 1.56 for $(\text{GeTe})_{0.85}(\text{PbSe})_{0.075}(\text{PbS})_{0.075}$ (Table R1), indicating the increase of lattice vibration anharmonicity.

Table R1. The elastic properties (Young’s modulus E , Poisson ratio r) and the Grüneisen parameter γ calculated based on the measured sound velocity.

Samples	v_L (m/s)	v_T (m/s)	v (m/s)	E (GPa)	r	γ
GeTe	3292	1938	2147	56.8	0.23	1.43
$(\text{GeTe})_{0.95}(\text{PbSe})_{0.025}(\text{PbS})_{0.025}$	3276	1911	2120	56.2	0.24	1.46
$(\text{GeTe})_{0.9}(\text{PbSe})_{0.05}(\text{PbS})_{0.05}$	3200	1840	2043	52.6	0.25	1.51
$(\text{GeTe})_{0.85}(\text{PbSe})_{0.075}(\text{PbS})_{0.075}$	3091	1755	1951	48.2	0.26	1.56

Revision: Page 11-12: “To prove the strengthened anharmonicity, Grüneisen parameters (γ) were calculated based on the sound velocities measured. Computational details and results are shown in the Supplementary Text and Table S1, respectively. The γ increases from 1.43 for GeTe to 1.56 for $(\text{GeTe})_{0.85}(\text{PbSe})_{0.075}(\text{PbS})_{0.075}$, indicating the increase of lattice vibration anharmonicity.”

4. In this manuscript, the symbols of both Seebeck coefficient and interaxial angles use α . In addition, the symbol of the Seebeck coefficient in Supplementary Note is S . Please

check for other errors.

Response: Thank you for your careful reading. We have amended these errors in the revised manuscript by using β to represent the inter-axial angles. The Seebeck coefficient (S) in Supplementary has been replaced by the symbol “ α ”.

5. In previous papers (Joule, 2018, 2, 976-987; Cell Reports Physical Science, 2022, 3, 101009), increasing interaxial angles is proved to be helpful in converging energy band in GeTe. In this manuscript, interaxial angles also increase with PbSe, PbS and Sb alloying/doping. Is there band convergence?

Response: Increasing interaxial angles can indeed promote band convergence in GeTe. We have calculated the band structures of GeTe with different interaxial angles, as shown in Fig. R3 and Fig. S23. The energy offset between multiple valence bands is reduced, which facilitates the transport of carriers through multiple bands. This results in the increase of density-of-states effective mass (m^*) with doping Pb, Se and S in GeTe (Fig. S22 and S27).

Fig. R3. Band structures of GeTe with different interaxial angles.

Revision: Page 11: “Moreover, the enhancement of α is also related to the increase of density-of-states effective mass (m^*) (Fig. S22), which is due to that PbSe and PbS alloying increases the interaxial angles of GeTe and facilitates band convergence, as demonstrated in Fig. S23.”

Reviewer #2 (Remarks to the Author):

This manuscript discusses an important problem in thermoelectric materials and beyond, specifically the mechanisms of doping or alloying. The authors propose that compounds with metavalent bonding easily mix to form solid solutions. The idea has been tested in the GeTe-PbSe-PbS system, which forms a single-phase solid solution, in contrast to GeTe-GeSe-GeS and GeTe-SnSe-SnS system. This idea is intriguing, and I recommend it to be published after addressing the following questions.

Response: We sincerely appreciate the positive evaluation of our work. We also thank the reviewer for the constructive comments and suggestions.

(1) For the difference between PbS alloyed GeTe and SnS/GeS alloyed ones, have the authors considered the different crystal structures of SnS and GeS? Typically, compounds with different crystal structures are harder to form a solid solution.

Response: This is a very nice comment. Indeed, the different crystal structures of these chalcogenides are determined by their different chemical bonding mechanisms. Metavalent bonding is characterized by a half-filled σ -bond constructed by the overlap of p-orbitals [Adv. Mater. 2023, 35, 2208485]. Due to the orthogonal alignment of p-orbitals, an ideal MVB solid should utilize a cubic structure. Yet, this configuration is energetically unstable, which spontaneously creates Peierls distortion to lower the energy of the system [J. Phys. D: Appl. Phys. 2020, 53, 234002]. Thus, rhombohedral GeTe is more stable under ambient conditions. Yet, the cubic structure can also be stabilized by increasing the charge transfer, which explains the rock-salt structure of PbSe and PbS. In striking contrast, the p-orbital overlap for GeS and SnS is much smaller than that in GeTe, leading to a significantly larger degree of Peierls distortion. This results in a structural phase transition from rhombohedral to orthorhombic associated with a chemical bonding transition from metavalent to covalent. In this regard, the low miscibility between materials with different crystal structures can also be ascribed to the different bonding mechanisms. In contrast, even though the crystal structures between rhombohedral GeTe and cubic PbS are different, they can still form a solid solution due to their same metavalent bonding mechanism.

Revision: We have added the above arguments to the revised manuscript on pages 8 and 9.

(2) PbS is a metavalent-bonding compound, whereas GeS and SnS are not. Therefore, it can be expected that in the local environment of S in GeTe-PbS system, Pb should be present. Is it possible that S-Pb pairs could form in this system.

Response: This is a thoughtful comment and question. We have no direct evidence to show the local S-Pb pairs. APT results just demonstrate a homogeneous distribution of all constituting elements. Yet, by comparing the different solubilities of S in GeTe depending on the presence of Pb, we can infer that Pb should occupy the nearest lattice sites of S. Otherwise, GeS precipitates will form with such a high nominal content of S (5%) in the matrix as observed in the other two cases without Pb. This also emphasizes the importance of local chemical bonding in stabilizing the crystal structure. Nevertheless, we prefer to avoid making such an argument in the manuscript since we do not have robust evidence to verify this assumption.

(3) The increased carrier concentration in GeTe-GeSe-GeS system has been attributed to the formation of GeS-rich secondary phases. However, GeS is not expected to consume Ge from GeTe matrix as it is added in the form of GeS, not S.

Response: Indeed, the formation of GeS precipitates should not change the content of Ge in the GeTe matrix according to the nominal stoichiometry of $(\text{GeTe})_{1-2x}(\text{GeSe})_x(\text{GeS})_x$. However, APT characterizations show that the compositions in both the GeTe-rich matrix and the GeS-rich precipitates are much more complex than their binary phase. Both Fig. 2c and Fig. S14 indicate that the content of Ge in the GeS-rich phase is higher than that in the GeTe-rich matrix. Thus, the experimentally observed enhancement of hole concentration with increasing x can be attributed to the increased content of Ge vacancies due to the formation of GeS-rich precipitates.

Revision: We have added one sentence on page 12 to explain the reason for the

increased Ge vacancy. “This could be due to the formation of GeS-rich secondary phases, which consume Ge from the GeTe matrix, *as can be indicated by the relatively higher content of Ge in the precipitates than that in the matrix determined by APT.*”.

(4) The decrease in lattice thermal conductivity in the GeTe-PbSe-PbS system could be caused by the stronger anharmonicity, which can be further testified by the higher Grüneisen constant. The author can calculate it through sound velocities.

Response: Thanks for your suggestion. We have calculated the Grüneisen parameters (γ) based on the measured sound velocity, as shown in Table R1. The γ increases from 1.43 for GeTe to 1.56 for $(\text{GeTe})_{0.85}(\text{PbSe})_{0.075}(\text{PbS})_{0.075}$, indicating the increase of lattice vibration anharmonicity.

Revision: We have added the above discussion on Pages 11-12: *“To prove the strengthened anharmonicity, Grüneisen parameters (γ) were calculated based on the sound velocities measured. Computational details and results are shown in the Supplementary Text and Table S1, respectively. The γ increases from 1.43 for GeTe to 1.56 for $(\text{GeTe})_{0.85}(\text{PbSe})_{0.075}(\text{PbS})_{0.075}$, indicating the increase of lattice vibration anharmonicity.”*

Reviewer #3 (Remarks to the Author):

This work proposed a doping strategy to tailor microstructures for metavalently bonded materials by understanding the relationship between the solubility of dopants and the chemical bonding mechanism. The authors demonstrated that high solubility can be achieved by alloying metavalently bonded solids. On the contrary, secondary phases are formed by alloying materials employing different chemical bonding mechanisms. This conclusion has been well demonstrated in a series of GeTe-based alloys. The high solubility of PbSe and PbS in GeTe is attributed to the metavalent bonding mechanism of these compounds. The generality has also been confirmed in SnTe-based alloys and applied to explain similar phenomena in other systems reported such as Bi₂Te₃ and

PbTe. As a result of the microstructure manipulation, the authors proved that a high solubility of PbSe and PbS in GeTe leads to an improved thermoelectric performance. This work broadens the understanding of microstructure and property manipulation by chemical doping. I am also very impressed by the beautiful data and figures. Therefore, I recommend the publication of this manuscript in Nature Communications. The following points should be addressed before the final publication.

Response: We appreciate the positive evaluation and insightful comments on our work.

1. In thermoelectrics, constructing a second phase is a common way to improve thermoelectric performance. Why does the second phase such as GeS not improve the thermoelectric properties of GeTe in this work?

Response: On one hand, the second phase should be in the nanoscale to improve the thermoelectric performance due to the short mean free path of phonons for typical thermoelectric materials. In contrast, the GeS-rich second phases observed in this work are on micrometer scales, indicating weak effects on scattering phonons. On the other hand, the thermoelectric performance of GeS with covalent bonding is lower than GeTe with metavalent bonding [Adv. Funct. Mater., 2020, 30, 1904862]. Therefore, the second phase of GeS does not improve the thermoelectric performance of GeTe in this work.

Revision: *Page 13: “Meanwhile, the GeS-rich second phase does not improve the thermoelectric performance of GeTe as well. On one hand, the more resistive second phase can only improve the overall thermoelectric performance if its size is comparable to the mean free path of phonons, which reduces the lattice thermal conductivity. Yet, the GeS-rich second phases observed in this work are on micrometer scales, indicating weak effects on scattering phonons in GeTe with an intrinsically small phonon mean-free path. On the other hand, the thermoelectric performance of GeS with covalent bonding is lower than GeTe with metavalent bonding⁸⁴. The composition of GeS into GeTe can hardly improve the overall thermoelectric performance.”*

2. The lattice thermal conductivity in the GeTe sample alloyed with PbSe and PbS is significantly reduced. Besides strong point defect scattering, the change of phonon dispersion can also lead to the reduction of thermal conductivity. What is the effect of Pb, Se, S doping or PbSe and PbS alloying on the phonon dispersion of GeTe?

Response: There are many possible unit-cell structures for the Pb, Se and S co-doped GeTe given its complex stoichiometry. We have constructed a special quasi-random structure (SQS) of $\text{Ge}_{48}\text{Pb}_6\text{Te}_{48}\text{Se}_3\text{S}_3$ using the Alloy Theoretic Automated Toolkit (ATAT) to simulate the disordered structure. Yet, due to the complexity of this structure, we could not finally obtain a reliable phonon dispersion of this compound.

In fact, the reduction of sound velocity after PbSe and PbS alloying indicates the softening of phonons (Fig. S25), which can be demonstrated by the increase of the Grüneisen parameters (γ) (Table R1). These results demonstrate that PbSe and PbS alloying not only enhances the point defect scattering but also softens phonons and increases the lattice vibration anharmonicity.

Revision: We have added the above discussion on Pages 11-12: *“To prove the strengthened anharmonicity, Grüneisen parameters (γ) were calculated based on the sound velocities measured. Computational details and results are shown in the Supplementary Text and Table S1, respectively. The γ increases from 1.43 for GeTe to 1.56 for $(\text{GeTe})_{0.85}(\text{PbSe})_{0.075}(\text{PbS})_{0.075}$, indicating the increase of lattice vibration anharmonicity.”*

3. Generally, doping-induced ionized impurity scattering will reduce carrier mobility. Why does Sb doping not significantly reduce carrier mobility?

Response: In general, the increased content of dopants could enhance the alloy scattering of electrons and thus reduce the carrier mobility. Yet, the reduction of carrier concentration of Sb doping weakens the scattering of electrons and then increases the carrier mobility. As a consequence, Sb doping does not significantly reduce carrier mobility.

Revision: Page 15: “The slight reduction of carrier mobility with increasing the content of Sb is ascribed to the compromise between the increased impurity scattering and the weakened carrier scattering. On one hand, the increased content of Sb dopants could enhance the alloy scattering of electrons and thus reduce the carrier mobility. On the other hand, the reduction of carrier concentration of Sb doping weakens the scattering of electrons and then increases the carrier mobility.”

4. There is a small diffraction peak in the 20-30° of XRD patterns (Fig. S1). What is that?

Response: The small diffraction peak at the 20-30° of the XRD pattern in Fig. S1 is indexed to the crystalline plane of Ge, i.e., there is a secondary Ge phase in all samples, which is a usual phenomenon in GeTe-based materials due to low formation energy of Ge vacancies. We have marked the diffraction peak in the XRD pattern, as shown in Fig. R4 and Fig. S1.

Fig. R4. Room-temperature XRD patterns of GeTe_{1-x}S_x samples.

5. There are a large number of domain structures in (GeTe)_{0.9}(PbSe)_{0.05}(PbS)_{0.05} sample (Fig. S15). What effect do they have on thermal transport property?

Response: Based on geometric phase analysis (GPA), there is a large strain in the boundary of the domain structure, as shown in Fig. R5 and Fig. S15. The strain can

effectively scatter heat-carrying phonons with low and intermediate frequencies [J. Am. Chem. Soc. 2024, 146, 12620-12635]. We have added the above discussions to the revised manuscript.

Fig. R5. **a** Transmission electron microscopy (TEM) image, **b** EDS mappings for $(\text{GeTe})_{0.9}(\text{PbSe})_{0.05}(\text{PbS})_{0.05}$ sample, **c** high-magnification TEM image of domain boundary, **d1-d4** geometric phase analysis (GPA) strain maps along the ϵ_{xx} , ϵ_{yy} , ϵ_{xy} , and ϵ_{rot} of **c**.

Revision: Page 7: “Geometric phase analysis (GPA) shows a large strain in the domain boundary, which can effectively scatter heat-carrying phonons with low and intermediate frequencies (Fig. S15c and 15d).”

6. As can be seen from Fig. S25, the m^* is increased with PbSe and PbS alloying, what are the effects of Pb, Se, and S doping on the band structure of GeTe?

Response: The interaxial angle of GeTe increases with increasing the alloying content of PbSe and PbS. We have calculated the band structures of GeTe with different interaxial angles, as shown in Fig. R3. We find that the energy offset between multiple valence bands decreases with increasing the interaxial angles. This results in the increase of m^* due to the band convergence effect (Fig. S22 and S27).

Revision: *Page 11: “Moreover, the enhancement of α is also related to the increase of m^* (Fig. S22), which is due to that PbSe and PbS alloying increases the interaxial angles and facilitates band convergence, as demonstrated in Fig. S23.”*

7. More details of the synthesis, characterization, and calculation should be provided (in the SI).

Response: Thank you for your suggestion. We have added more details about the synthesis, characterization, and calculation in the Supplementary Information.

8. Some newly reported papers about GeTe should be mentioned in the manuscript, e.g., Adv. Energy Mater. 2024, 14 (26), 2400340; J. Am. Chem. Soc. 2024, 146 (2), 1681; Adv. Funct. Mater. 2023, 33 (25), 2301750.

Response: We agree that adding recently reported papers can help better understand the GeTe-based materials. We have added these papers in the “Introduction” part.

REVIEWERS' COMMENTS

Reviewer #1 (Remarks to the Author):

All reviewer comments are well addressed, so I'd like to recommend this manuscript to be accepted by Nature Communications.

Reviewer #2 (Remarks to the Author):

The authors have well addressed my questions. Therefore, I recommend its acceptance.

Reviewer #3 (Remarks to the Author):

All the reviewers' comments have been addressed adequately, therefore I'd like to recommend this manuscript to be published in NC.

Response to Reviewers

Reviewer #1 (Remarks to the Author):

All reviewer comments are well addressed, so I'd like to recommend this manuscript to be accepted by Nature Communications.

Response: Thank you so much.

Reviewer #2 (Remarks to the Author):

The authors have well addressed my questions. Therefore, I recommend its acceptance.

Response: Thanks a lot.

Reviewer #3 (Remarks to the Author):

All the reviewers' comments have be addressed adequately, therefore I'd like to recommend this manuscript to be published in NC.

Response: Many thanks.